# (GIGA)bYte

TECHNICAL RELEASE

# Building a community-driven bioinformatics platform to facilitate *Cannabis sativa* multi-omics research

Locedie Mansueto[1], Tobias Kretzschmar[1], Ramil Mauleon[1,2,*] and Graham J. King[1,3]

1 Southern Cross University, Military Road, Lismore New South Wales, 2480, Australia
2 International Rice Research Institute, Pili Drive, Los Baños Laguna, 4031, Philippines
3 Recombics, Alstonville, New South Wales, 2480, Australia

## ABSTRACT

Global changes in cannabis legislation after decades of stringent regulation and heightened demand for its industrial and medicinal applications have spurred recent genetic and genomics research. An international research community emerged and identified the need for a web portal to host cannabis-specific datasets that seamlessly integrates multiple data sources and serves omics-type analyses, fostering information sharing.

The Tripal platform was used to host public genome assemblies, gene annotations, quantitative trait loci and genetic maps, gene and protein expression data, metabolic profiles and their sample attributes. Single nucleotide polymorphisms were called using public resequencing datasets on three genomes. Additional applications, such as SNP-Seek and MapManJS, were embedded into Tripal. A multi-omics data integration web-service Application Programming Interface (API), developed on top of existing Tripal modules, returns generic tables of samples, properties and values. Use cases demonstrate the API's utility for various omics analyses, enabling researchers to perform multi-omics analyses efficiently.

**Availability and implementation:** The web portal can be accessed at www.icgrc.info.

**Subjects** Software and Workflows, Bioinformatics, Agriculture

**Submitted:** 18 June 2024

\* Corresponding author. E-mail: ramil.mauleon@scu.edu.au

Preprint submitted at https://doi.org/10.1101/2024.10.02.616368

## STATEMENT OF NEED

### Background on cannabis

*Cannabis sativa* (cannabis) is one of the earliest domesticated crops for food, fiber and medicine [1]. As hemp, cannabis was a significant source of fiber in ropes traditionally used for shelter, clothing, traps, harnesses, sails and halyards. Its seeds (hempseed) are relatively rich in nutritious oils of high omega-3 and omega-6 fatty acid content [2], proteins, and a variety of micronutrients for the human diet [3]. Its female flowers produce a mixture of secondary metabolites, some of which have psychotropic effects and have been used in religious rituals and recreational use. Because of these properties and uses, cannabis was also regarded as an ancient 'wonder plant' coined with various polarizing names, including 'devil's weed', 'penicillin of Ayurvedic medicine', and 'donator of joy' [4]. The secondary metabolites, which are highly abundant in glandular trichomes on the surface of female flowers for pathogen resistance [5], insect deterrent [6], and UV protection [7], also have

pharmacological and nutritional potential [8]. Cannabis glandular trichomes produce large amounts of terpenoids and cannabinoids. Calvi *et al.* [9] identified 554 compounds in cannabis, including 113 cannabinoids and 120 terpenes. More than 200 metabolites from different genotypes are volatile, with 58 monoterpenes and 38 sesquiterpenes being reported. The most abundant monoterpenes are limonene, b-myrcene, a-pinene and linalool, while sesquiterpenes are E-caryophyllene, b-farnesene and b-caryophyllene.

Cannabis is dioecious and diploid, with nine autosomes and a pair of sex chromosomes (female XX, male XY). The diploid genome size estimated by flow cytometry is 1,636 Mb for female plants (818 Mb haploid size) and 1,683 Mb (haploid size 843 Mb) for male plants [10]. The first draft sequence reported a haploid size of 534 Mb [11], while the accepted standard reference genome to date is 876 Mb [12].

Cannabis had been classified as a narcotic for most of the 20th century [13]. However, recent revisions of its legal status and the recognition of defined medicinal benefits have supported cannabis as an emerging economic crop with the scheduling recommendation of the World Health Organization [14]. Improving cannabis for its food, fiber and medicinal traits has thus recently been the focus of both public and private sector research [15]. The global market projection for medicinal cannabis is $176 billion by 2030 [16]. At a 21% cumulative growth rate, the global cannabis cultivation market, including hemp, could reach $1.8T by 2030 [17].

## The need for a cannabis multi-omics database and analyses platform

The assembly of reference genomes [12, 18–21] for hemp and medicinal types has facilitated an abundance of publications and their associated publicly available genomic, transcriptomic, proteomic, and metabolomic datasets. Data available from general web portals also include cannabis genomic and other omics data (e.g., NCBI, ENA, DDBJ, CoGe) [22–24], plant-specific databases (e.g., EnsemblPlant, Phytozome, PlantCyc) [25–28] and Cannabis-specific databases. Kannapedia [29] is a commercial site from Medicinal Genomics, which provides summaries on the genomic variation of cannabinoid biosynthesis genes for nearly two thousand cultivars. Bulk downloads of most of the data, however, are not available as a built-in feature. CannabisDraftMap.org [30] provides an inventory of raw proteomics and metabolomics data, but the site has no analysis tools nor processed results available. The Cannabis Genome Browser [31] hosts the results from the first cannabis draft sequence publication [11] but has not been updated since. Leafly [32] is a commercial site with cannabinoids and qualitative descriptions of terpenes. SeedFinder [33] provides an interface for cannabis breeders, seedbanks, growers and enthusiasts by collecting and standardizing information about cannabis. As of February 2024, the database has 31,823 different cultivars listed. While it is not an online shopping platform, it is supported by related advertisements. CannabisGDB [34] is an integrated cannabis functional genomics database with four modules: varieties, gene loci, metabolites and proteins. 'Varieties' provides descriptions of eight assembled genomes that are available from NCBI. 'Gene loci' contains the in-house gene predictions and functional annotation on these eight genomes, and these are displayed using JBrowse (RRID:SCR_001004) [35]. 'Proteins' includes the UniProt plant proteins matched to the in-house protein model. 'Metabolite' displays various bar charts showing the percentage of cannabinoid content collected from external publications.

Most of the cannabis-specific websites and databases described above host proprietary data; hence, the focus is mainly on specific cultivars and trait characterization. Some have omics-type data but are usually from unpublished or proprietary sources. These limitations motivated this project to build an authoritative 'open-science' reference bioinformatics portal to support cannabis multi-omics research with capabilities that we believe will significantly benefit the research community, including (a) tools to perform quick analyses using the loaded datasets, (b) tools allowing data/contents curation by the community, (c) creation and promotion on the use of research data standards, and (d) a platform for the collaboration and communication by the cannabis research community.

The features and capabilities were initially intended to fulfill the needs defined by the International *Cannabis* Genomics Research Consortium (ICGRC). Formed during the 2020 Plant and Animal Genome conference by an international community of cannabis researchers from the public and private/commercial sectors, ICGRC aims to provide an authoritative source of rigorous and peer-reviewed scientific information relating to Cannabis genetics, genomics and phenotyping. The scope of ICGRC activities encompasses (a) information and data sharing, (b) establishing standards and reference information, (c) facilitating the unambiguous description and availability of genetic resources, (d) contributions to the consistent annotation of gene identity and function, and (e) describing common genetic resources and, where necessary, virtual gene-banks [36].

## IMPLEMENTATION

### Tripal was the chosen technology for the cannabis omics platform

There is a range of widely-adopted systems in the public domain suitable for genomic and genetic web databases/portals, such as InterMine (RRID:SCR_001772) [37], MetaCyc (RRID:SCR_007778) [38], Ensembl Plants (RRID:SCR_007778) [26], and Tripal [39]. Tripal is a content management system for genomics data based on the Drupal platform [40]. The data architecture is based on Chado (RRID:SCR_024073) [41], a highly normalized schema that uses biological ontologies to define bioentities and their relationships. Tripal architecture is modular, allowing plug-in features, such as ready-to-use specialized 'omics modules for loading, visualizing and analyzing, built-in support for user-driven content management, and is GMOD [42] standards-compliant. Tripal is used in many plant genomic research communities [39, 43] and various genome hub projects [44], including several from the SouthGreen bioinformatics platform [45]. Its major advantage being flexibility for community-wide contribution and content curation, Tripal was chosen as the software platform for the project.

The Tripal setup for this project, referred to as protocol from here onwards, is described in the ICGRC Portal Tripal Data Generation and Setup in protocols.io [46]. The main protocol steps are: cannabis dataset download, reanalysis and formatting for Tripal loading (protocol steps 1-8), Tripal docker container instantiation (protocol step 9), and loading of the reference genomes and annotation (protocol step 10). Modules have then been installed, enabled and loaded with the corresponding data: gene_search (protocol step 11), expression (protocol step 12), phenotype (protocol step 13), map (protocol step 14) and synteny (protocol step 15).

## Cannabis omics, genetic and phenotypic datasets in the public domain

Published omics datasets are sourced from databases such as NCBI Sequence Read Archive (SRA) (RRID:SCR_004891) (raw sequences) [47], NCBI Reference Sequence Database (RefSeq) (RRID:SCR_003496) (genome assemblies and annotation) [48], Gene Expression Omnibus (GEO) (RRID:SCR_005012) (gene expression) [49], UniProt (RRID:SCR_002380) (protein sequences and annotations) [50], OBO Foundry biological ontologies and controlled vocabulary OBO (RRID:SCR_007083) [51], and Crop Ontology (RRID:SCR_010299) [52] crop-specific controlled vocabulary. Datasets have also been manually curated from the published article tables, figures, and supplementary data and sites, which are common sources, especially for non-model or less studied species like cannabis.

### *Genome assemblies and annotations*

The first peer-reviewed publication for cannabis genomes was on Purple Kush (PK), a medicinal type tetrahydrocannabinol (THC)-dominant marijuana cultivar, and Finola (FN), a hemp-type cultivar developed for oil seeds [11]. Both PK and FN assemblies were updated and improved in Laverty *et al.* [18]. CBDRx (cs10) is a cannabidiol (CBD)-type cultivar derived from a cross of Skunk#1 (marijuana) and Carmen (hemp), and is estimated to be 89% marijuana and 11% hemp by admixture analysis. This was the first cannabis assembly with NCBI RefSeq gene annotations (cs10, GCF_900626175.2) for three years and was used by many publications for gene assignment. Pink Pepper (ASM2916894v1) is the second assembly annotated by RefSeq recently (November 2023).

Three primary genome assemblies are currently loaded to the portal: cs10 (also known as CBDRx, a CBD-dominant line) [12], PK and FN [18]. They were chosen to represent the major cannabis end-uses and have transcript sequences for gene annotation. Other cannabis genome assemblies currently in the public domain, such as the cultivars First Light [12], Jamaican Lions [21], Cannbio-2 [20], JL wild [19], and Pink Pepper, were not included in this release of the portal.

The annotations from RefSeq for cs10 and our FINDER [53] annotation for PK and FN were loaded and are searchable using the 'Gene Function Search' module interface, and are displayed in the JBrowse browser. The query constraints in the functional annotations are cultivar/assembly, annotation analysis, genome region, gene or transcription accessions, and keyword. Query by sequence is also possible using the BLAST interface (RRID:SCR_004870). A summary of annotations available in the portal is in Table 1.

### *Transcriptomes, comparative expression, and proteomics datasets*

The public availability of transcript data has been the primary consideration for our choice of genomes to load in the portal (Table 2). For comparative analyses, either absolute expression levels across multiple samples or relative fold-change of transcript levels between two samples are available. Currently, three (3) Cannabis transcript assemblies are publicly available: canSat3, finola1 and cannbio2. All were used to annotate their respective genome and in differential expression studies across tissues [11, 20, 54]. These transcript assemblies are viewable in the JBrowse browser. Functional annotation was assigned for each transcript according to homology with RefSeq mRNA sequences, and the genome location by splice-aware alignment with different genomes. As with gene loci, the transcript



**Table 1.** Gene annotations available for various published cannabis genomes.

| Cultivar | Annotation | Type | Genome assembly | RNA-Seq/transcripts dataset | Method | Reference |
|---|---|---|---|---|---|---|
| *Cannabis sativa* (CBDRx, cs10) | CS10 | gene, mRNA | GCF_900626175.2 | | NCBI RefSeq | RefSeq |
| *Cannabis sativa* (Purple Kush, pkv5) | PKGMv1 | gene, mRNA | GCA_000230575.5 | | GeneMark EP gene prediction, best hit blastp with NCBI NR | [11] |
| *Cannabis sativa* (Purple Kush, pkv5) | PKFDv1 | gene, mRNA | GCA_000230575.5 | PRJNA73819 (SRR352195, SRR352196, SRR352198, SRR352200, SRR352201, SRR352202, SRR352203, SRR352205, SRR352208, SRR352210) | FINDER This study (protocol steps 2–3) | This study [11] |
| *Cannabis sativa* (Finola, fnv2) | FNFDv1 | gene, mRNA | GCA_003417725.2 | PRJNA73819 (SRR351932, SRR351933) PRJNA483805 (SRR7630400-SRR7630408) | FINDER This study (protocol steps 2–3) | This study [11, 55] |
| (CBDRx, cs10) | cs10g21t | gene, mRNA | GCF_900626175.2 | 21 Trichomes | This study (protocol step 4) | This study [54–57] |
| *Cannabis sativa* (Purple Kush, pkv5) | Csativa | mRNA | | Assembled transcripts from canSat3_transcriptome-representative.fa.gz | | [11] |
| *Cannabis sativa* (Finola, fnv2) | Csativa | mRNA | | Assembled transcripts from finola1_transcriptome-representative.fa.gz | | [11] |
| *Cannabis sativa L.* | Csativa | mRNA | No specific assembly | C sativa mRNAs from NCBI Genes | | |

assemblies can be queried on the 'Gene Function Search' page, and their locations and transcript levels are also available to be displayed in JBrowse.

Comparative expression studies for cannabis are limited and mostly explore changes relative to secondary metabolism and fiber synthesis, comparing different cultivars, treatment conditions, tissues or developmental stages. A gene expression study for bast fiber development in the textile hemp variety Santhica 27 included tissues sampled from the bottom, mid and top stem regions [58]. In another study, FN trichomes were separated from mid-stage flowers and categorized into bulbous, sessile and mature stalks [55]. Terpene and cannabinoid profiles were measured, and the correlation with synthase gene transcript levels was assessed [56, 57]. Relevant expression datasets in NCBI GEO were also loaded. The list of absolute and differential expression datasets available within the ICGRC Tripal resource is shown in Tables 3 and 4, respectively. Expression data are displayed as transcript heatmaps of genes by samples. It is important for this module to have traceable assignment of sample attributes from the source. Expression data are also displayed overlaid on the generic Biopathways module.

At the completion of this project, only one cannabis proteome expression dataset had been made available in GEO. Using quantitative time of flight mass spectrometry, 1,240 proteins from glandular trichome heads, 396 from trichome stalks, and 1,682 from whole flower tissue isolates were identified by Conneely *et al.* [59].

### Variants discovery using resequencing datasets

Details on the reanalysis of Next Generation Sequencing (NGS) datasets of cannabis are described in an accompanying paper [60] about the CannSeek single nucleotide polymorphisms (SNPs) database (RRID:SCR_025579) [61]. Data available in NCBI SRA as of December 2022 consisting of 26 trichome RNA sequencing (RNA-Seq), 383 genomic DNA, and the Phylos amplicons were reanalyzed using GATK (RRID:SCR_001876) [62] and Parabricks [63] variant calling pipelines. The results are displayed using the CannSeek



**Table 2.** Transcript assemblies available from published cannabis studies.

| Dataset | Method | Samples | Datasource | Reference |
|---|---|---|---|---|
| canSat3<br>finola1<br>Cannbio-2 | Aligned with cs10 and pkv5 assemblies using minimap2 in splice alignment mode | Purple Kush<br>Finola<br>Cannbio-2 | canSat3_transcriptome-representative.fa.gz<br>finola1_transcriptome-representative.fa.gz<br>GIFP01000001:GIFP01064413 | [11]<br>[11]<br>[54] |
| 5 transcripts | 1. Clustered (mmseq2 cm2 80% identity) transcript assemblies combined from 5 data sources<br>2. Aligned to cs10 and pkv5 assemblies using minimap2 in splice alignment mode | Purple Kush<br>Finola<br>Cannbio-2 | canSat3_transcriptome-full.fa.gz<br>finola1_transcriptome-full.fa.gz<br>GIFP01000001:GIFP01064413 | [11]<br>[11]<br>[54] |
| | | Terpene synthase genes from various *C. sativa* cultivars | AB057805.1, AB164375.1, AB164375.1, AB292682.1, CcTHCAs, CsCBDAs, KY014554.1, KY014555.1, KY014556.1, KY014557.1, KY014558.2, KY014559.1, KY014560.1, KY014561.1, KY014562.1, KY014563.1, KY014564.1, KY014565.1, MK131289.1, MK131290.1, MK614217.1, MK801762.1, MK801763.1, MK801764.1, MK801765.1, MK801766.1, MN967470.1, MN967471.1, MN967472.1, MN967473.1, MN967474.1, MN967475.1, MN967477.1, MN967479.1, MN967480.1, MN967481.1, MN967481.1, MN967482.2, MN967483.1, MN967484.1, MnCBDAs-like, MT295505.1, MT295506.1, TPS11, TPS11-like, TPS12, TPS12-like, TPS13-like1, TPS13-like2, TPS14, TPS17, TPS20, TPS23, TPS24, TPS30-like, TPS32, TPS36, TPS37, TPS38, TPS40, TPS41, TPS42, TPS43, TPS44, TPS46, TPS47, TPS48, TPS49, TPS4-like, TPS50, TPS51, TPS52, TPS53, TPS55, TPS58, TPS59, TPS60, TPS61, TPS62, TPS63, TPS64, TPS6-like, TPS7_Finola, TPS8-like, TPS9, TPS9-like1, TPS9-like2 | [56, 57, 64] |
| | | Cannabinoid synthase genes from Argvana Heart and Tachllta cultivars | GHVF01000001-GHVF01000012, GHVG01000001-GHVG01000012 | [65] |
| 21 trichomes samples | This study (protocol step 4)<br>1. Fastq raw reads from trichome RNA-Seq experiments were assembled using rnaspades<br>2. Transcript assemblies were aligned to cs10 and pkv5 assemblies using minimap2 in splice alignment mode | Sour Diesel, Canna Tsu, Black Lime, Valley Fire, White Cookies, Mama Thai, Terple, Cherry Chem, Blackberry Kush | PRJNA498707 | [57] |
| | | Afghan Kush, Blue Cheese, CBD Skunk Haze, Chocolope, Lemon Skunk | PRJNA599437 | [56] |
| | | Finola bulbous, sesille, stalked | PRJNA483805 | [55] |
| | | Cannbio-2 trichome, 4 development stages | SRR10600922, SRR10600916, SRR10600908, SRR10600906 | [54] |

**Table 3.** Absolute gene/transcript expression data from published cannabis studies.

| Samples | Tissue/condition | Treatment | Measurements | Reference |
|---|---|---|---|---|
| 21 Trichome samples | 21 Trichome | none | Cs10 mRNA | This study (protocols.io step 5) |
| 21 Trichome samples | 21 Trichome | none | cs10g21t | This study (protocols.io step 4–5) |
| *C. sativa* cv. Santhica 27 | Three stem regions: top, middle, bottom | none | FN, finola1 | [58] GEO GSE94156 |
| Purple Kush, Finola, USO-31<br>SRP006678<br>SRP008673 | SRP006678 mature leaf, immature leaf, flower buds, primary stem, young leaf, mature flower, stem-petioles, entire root<br>SRP008673 young leaf, roots, shoots, stems, pre-flowers, early-stage flowers, mid-stage flowers | none | PK, canSat3 | [66] GEO GSE93201 |
| Hemp cultivar Yunma 1 | - leaves, roots, stem bast, and stem shoots were collected separately.<br>- Separate ABA treatment and collected after 3, 6 hrs<br>- Equal samples pooled | drought-stressed plants (DS), control (CK), abscisic acid | PK, reads aligned to Purple Kush AGQN00000000 and transcript assembly with Cufflinks | [67] GEO GSE56964 |
| Finola | Trichome: bulbous, sessile, stalked | none | FN, finola1 | [55] |

**Table 4.** Differential gene/transcript expression data from published cannabis studies.

| Samples | Tissue | Measurements | Reference |
|---------|--------|--------------|-----------|
| Cannbio-2 | - stem, root-tip, root-mid, leaf from freshly planted cutting, vegetative plant to reproductive plant.<br>- floral bud tissues and trichomes were isolated from reproductive plants at 35, 42, 49 and 56 days after induction of flowering in the female plants<br>- male vegetative leaf and reproductive tissues (pollen sacs) from the male strain | Cannbio2 TSA<br>GIFP00000000 | [54]<br>PRJNA560453 |
| Hemp No. 8 | Flower, glandular trichome head and stalk | cs10 peptide, hemp on uniport | [59] |

software, derived from Rice SNP-Seek [68], hosted in a Podman or Docker container [69, 70] and embedded as a Tripal page.

### Maps

The interactive *Map Viewer*, a part of the Tripal platform [71], can display an entire genome or chromosome, and features such as markers and quantitative trait loci (QTLs) in a single view. Maps positions may either be physical (using base position, or bp) or genetic (using centiMorgan, or cM). Common markers are linked between the genetic and physical maps when displayed side by side. Public cannabis genetic linkage data are scarce; only two recent publications of QTL maps are currently loaded: a study on gene duplication and divergence affecting drug content [72] and QTL mapping for agronomic and biochemical traits [73].

### Phenotypes

The *Phenotype Viewer* displays the distribution of measured traits, including metabolites (cannabinoid and terpenoid content and their relative composition) across samples and experiments, visualized in a 'violin plot' [74]. Table 5 lists the phenotypic datasets loaded. This feature requires three genus-specific controlled vocabularies for trait, method and unit. During setup, they can be defined from existing ontologies or community customized. New terms may be added as new datasets are loaded. Currently, these ontologies use ChEBI [75] for chemical traits, PECO [76] for method and UO [77] for units. In the absence of a cannabis-specific vocabulary for methods and units in Crop Ontology, publication-specific terms were created when no suitable matches in the existing ontologies were found.

## Preparing datasets for the Tripal modules
### Gene model prediction and annotations

All resequencing data were trimmed using Trimmomatic (RRID:SCR_011848) [78] (protocol step 1). Cs10 genes and annotations were downloaded from NCBI RefSeq (protocol step 2). The FINDER [53] pipeline was used to predict gene models for updated assemblies of PK and FN using cultivar-specific RNA-Seq data. GffRead (RRID:SCR_018965) [79] was used to cluster overlapping gene loci predictions into a common locus (protocol step 3). Proteins were annotated for putative function/structure/location by homology search against the UniProt [50] databases using MMseqs2 (RRID:SCR_022962) [80]. Gene Ontology (RRID:SCR_002811) and InterPro (IPR) (RRID:SCR_006695) protein domain annotation of these proteins were done using InterProScan (RRID:SCR_005829) [81].

**Table 5.** Phenotype data from published cannabis studies.

| Reference | Cultivars | Tissue | Measurements |
|---|---|---|---|
| [82] | (33 × 3 reps) accessions sourced from the Ecofibre | leaf | Delta9-tetrahydrocannabinol, cannabidiol, cannabigerol |
| [83] | (30) Alien Blues, Blue Cherry Pie, Blue Dream, Bohdi Tree, Cold Creek Kush, Crystal Cookies, Double Royal Kush, FLO, Grandaddy Purple, Green Crack 2015, Green Crack 2017, Holy Power, Juanita, Lavendar Jones 2015, Lavendar Jones 2017, Lavendar, Love Lace, Oracle, Platinum Buffalo, Platinum Gorilla, Platinum Scout, Purple Fat Pie, Romulin, RX, Skywalker, Sour Willie, Thunderstruck, Twisted Velvet, Walker Kush, White Widow | flower, leaf | Delta9-tetrahydrocannabinol, cannabidiol, cannabichromene, cannabigerol, Delta9-tetrahydrocannabivarin |
| [57] | (9) Sour Diesel, Canna Tsu, Black Lime, Valley Fire, White Cookies, Mama Thai, Terple, Cherry Chem, Blackberry Kush | bud | (27) Delta9-tetrahydrocannabinolic acid, Delta9-tetrahydrocannabinol, cannabidiolic acid, cannabidiol, cannabigerol, cannabinol, cannabichromene, Total_cannabinoids, beta-myrcene, limonene, alpha-pinene, beta-pinene, 1,8-cineole, linalool, terpinolene, borneol, beta-ocimene, camphene, gamma-3-carene, camphor, plus-terpinene, Total_monoterpenes, beta-caryophyllene, alpha-humulene, nerolidol, Total_sesquiterpenes, Total_terpenoids |
| [56] | Lemon Skunk<br>Jack Herer<br>Blueberry<br>Chocolope<br>Afghan Kush<br>CBD Skunk Haze<br>Vanilla Kush<br>Blue Cheese | flower | (47) $\alpha$-pinene, myrcene, green-leaf-volatile, limonene, (E)-$\beta$-ocimene, linalool, terpinolene, Monoterpene-1, Monoterpene-alcohol, bergamotene, bisabolane-1, $\beta$-caryophyllene, $\gamma$-elemene, sesquiterpene-1, (E)-$\beta$-farnesene, $\alpha$-humulene, cadinane-1, cadinane-2, sesquiterpene, B-eudesmene, a-guaiene, bisabolane-2, sesquiterpene-3, sesquiterpene-alcohol-1, himachalane, sesquiterpene-4, unknown-compound, Guaiane-1, eudesma-3,7(11)-diene, $\alpha$-farnesene, sesquiterpene-5, Guaiane-2, sesquiterpene-6, caryophyllene-oxide, guaiol, cedrol, sesquiterpene-alcohol-2, sesquiterpene-7, cadinane, unknown-compound, sesquiterpene-alcohol-3, Guaiane-4, Eremophilane, bisabolol, selinane, guaiane-5, bulnesol |
| [84] | 35 from Ecofibre, 48 from IPK_CAN | | (12) cannabidiol, cannabidiolic acid, Delta9-tetrahydrocannabinol, Delta9-tetrahydrocannabinolic acid, cannabigerol, cannabigerolic acid, cannabidivarin, cannabidivarinic acid, tetrahydrocannabivarin, tetrahydrocannabivarinic acid, cannabinol, cannabichromene |

### Gene expression

Expression data from GEO [49] were loaded using the loader for expression and biosample. To build a system that hosts and integrates multiple omics datasets for the same sample sets, secondary analyses were performed on public data to complete these for all omics data types. Metabolite measurements were loaded and RNA-Seq sequences were reanalyzed. Gene expression using cs10 mRNAs was quantified on 21 trichome-specific public RNA-Seq datasets [54, 55, 57, 85] using Salmon (RRID:SCR_017036) (protocol step 5). Transcripts from the 21-trichomes RNA-Seq samples were assembled following (protocol step 4) using rnaSPAdes (RRID:SCR_016992) [86], then mapped to the three references using GMAP (RRID:SCR_008992) [87], and ORFs were identified using TransDecoder (RRID:SCR_017647) [88]. Quantification of transcript abundance was then performed using Salmon (RRID:SCR_017036) following (protocol step 5).

### Synteny block discovery

This follows the Tripal synteny viewer instructions for generating synteny blocks using gene annotations (protocol step 8). BLASTN (RRID:SCR_001598) and MCScanX (RRID:SCR_022067) [89] were used on the cs10 gene annotation and our generated predictions for PK and FN.

### BLAST database

BLAST databases for the three genomes and the annotated genes are installed in the portal, allowing genes and protein sequences from the users to be searched against genome assembly, coding sequence and proteins for cs10, PK, and FN. Cs10 gene model sequences are from NCBI Refseq, while PK and FN are from the FINDER gene prediction pipelines.

## Data preparation for external, non-Tripal tools

Some data are hosted and visualized using modules that are not part of Tripal (e.g., genome browser, pathways visualization, genotype management). Although Tripal might have some equivalent module, or the module might be implemented differently in other Tripal sites, the following tools were used for user benefit.

### JBrowse Genome Browser

The JBrowse Genome Browser [35] displays genome features along the reference chromosome or scaffolds. The data is loaded from interval files like General Feature Format (GFF) or Browser Extensible Data (BED) format. It is highly interactive and web-based, making it popular and used by many genomic database sites. Genes and transcript alignments for cs10, PK and FN genomes are available.

### Biopathways mapping

MapManJS, a web version of MapMan (RRID:SCR_003543) [90], is a gene expression and biological pathway visualization tool. It uses curated, organism-specific sets of genes for each pathway (protocol step 7). Since MapMan does not have cannabis-specific pathways, we lifted over (via best match with cannabis cs10 genes using mmseq2 reciprocal best hit on proteins) genes and pathways from other plant maps, specifically *A. thaliana*, tomato and eucalyptus. The expression of the cs10 genes is visualized over the selected pathway mapping, and the study is loaded into the Expression module.

### SNPs discovery and genotype data management

Variant data of the 21 trichomes RNA seq were discovered using the GATK-RNASeq (RRID:SCR_001876) pipeline (protocol step 6). Genomic resequencing data for 398 samples were also used to discover SNP variants using the GATK germline SNP calling workflow [91] and Parabricks [63].

 Genotype data results from variant discovery pipelines are presented as a large genotype matrix of samples and base position. The Tripal Natural Diversity Genotypes [92] module used in KnowPulse [74] is able to display this matrix. However, it uses web technologies that limit its interactivity and has performance issues on larger genotype matrix queries. The Rice SNP-Seek database [68] can display thousands of SNP positions and samples while maintaining interactivity, allows more analyses of the result and was thus used in this project.

## RESULTS

## Setup and customizations of the Tripal platform, and re-engineering modules for multi-omics data integration

The detailed workflow to set up and customize the platform is enumerated in the protocol [46] and illustrated in Figure 1. Documentations about instantiating various Tripal



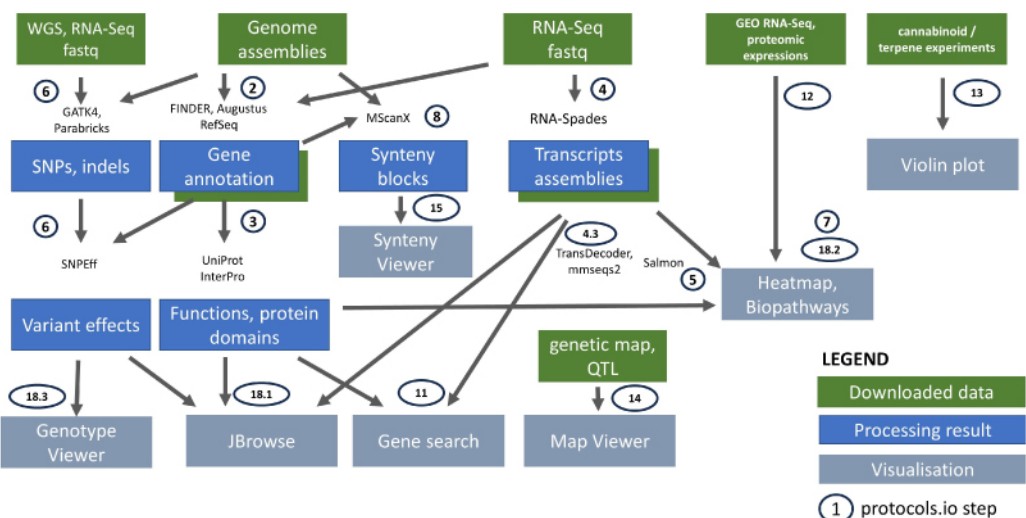

**Figure 1.** Workflow for the preparation of publicly available and reanalyzed data, for the analysis tools, and visualization features in the ICGRC Cannabis Web Portal (icgrc.info).
'Downloaded data' are datasets downloaded either from public databases, supplementary files, or manually curated from articles. 'Processing result' are the new datasets generated by this project, and value added from the legacy results of the publications. 'Visualization' is the Tripal modules and specialized web applications included in the portal for the interactive user interface. Implementation details are documented in [46].

modules and loading data are notoriously scattered across publications, and this workflow aims to collate and summarize the steps required to format files to simplify the preparation of the various modules. The workflow generates the files necessary to initially populate and test the different Tripal modules. This can also guide installations of future Tripal sites, especially for plants with scarce published data, or where only raw sequences are available.

We added features that are not part of Tripal and that were essential to enable multi-omics analyses, including pathway and expression visualization (MapMan JS) and genotype mining and visualization (SNP-Seek). NCBI BLAST was also included in the platform. protocol step 18 provides details on the installation of these external applications (JBrowse, CannSeek, MapManJS, BLAST).

In addition, we developed a novel module specifically for multi-omics integration. This is in the form of a web service with a predefined Application Programming Interface (API) that queries and merges the different Tripal modules, facilitating rapid retrieval of different data types from multiple sources. The capabilities of this API are demonstrated with use cases typically performed in omics studies.

### *Challenges encountered in the existing Tripal and Chado design and solutions implemented*

Although existing Tripal modules can host and display various biological omics datasets, integrating data from the different modules is not straightforward. The underlying Chado schema is highly flexible and normalized. Unlike the more common Entity-Relation, Chado follows the Entity-Attribute-Value Model database model [93]. The data model relies on biological ontologies, which are loaded as data, to define datatypes and relationships. The different loaders for each module populate database tables, which may be scattered, duplicated and confusing in their utility and terminologies.

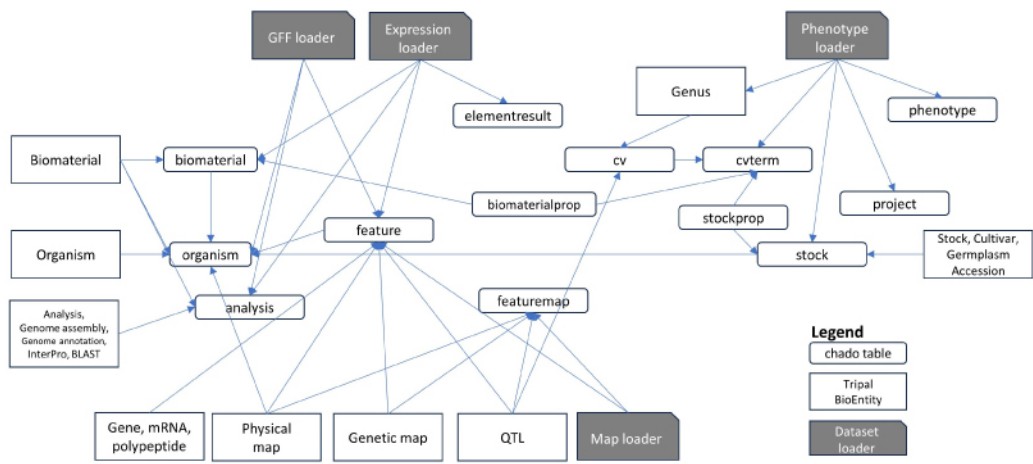

**Figure 2. Mapping of Tripal BioEntities and Chado tables.**
Input files are loaded using 'Dataset loader' for the genomic features GFF, expression, phenotype, or map datasets.
Input files are loaded using 'Dataset loader' for the genomic features GFF, expression, phenotype, or map datasets.
The fields are stored in various tables in the Chado schema (green). Tripal then defines BioEntities (blue) based
on the record types (type_id) in the tables. The type_id's are biological ontology terms loaded in the cvterm and cv
tables.

An additional complication is that Tripal introduces a layer of abstraction for biological entities on top of the storage tables in the Chado schema. Using ontology terms as type identifiers, a Chado table can store records for different biological entity types. Table 6 lists the default Tripal entities and their corresponding Chado tables and types. The Tripal BioEntity objects and Chado relational table mappings are illustrated in Figure 2. The feature, analysis and stock tables are used in common by multiple Tripal BioEntity types. Data management can be challenging when mapping the relationships between distinct bioentities because they are not equivalent to the entity-relationship defined by the Chado schema. Another challenge is the requirement to manually synchronize the Chado records and the Tripal entities when loaders are not used. There are synchronization buttons in Tripal to publish Chado records to Tripal, unpublish deleted records, or delete unpublished bioentities. Using these features requires a thorough understanding and experience with the datasets and platform.

Several solutions are implemented to overcome the challenges encountered. The integration process is built on top of existing Tripal modules, specifically for the 'chado_gene_search', 'expression' and 'phenotype' modules. These modules generate materialized views from complex queries used in the visualizations. The omics integration API queries these materialized views using SQL. Merging from the different modules is implemented using table pivot. All queries return a triplet of property, biomaterial and value. A pivot function then transforms them into a table with the biomaterials as column labels and properties as row labels. Figure 3 illustrates how the modules are linked for integration. Biomaterial labels can be from 'dbxref', 'biomaterial' or 'stock' tables. Properties are from 'cvterm' (sample metadata, traits) or 'feature' (genes, transcripts, proteins). The values are collected from any of the other tables following their references. Controlled terms are defined in the 'cvterm' table, but their assignment to a specific biomaterial and property are in their datatype-specific tables. The generic omics query

**Table 6.** Tripal objects and CHADO tables utilized in the ICGRC portal.

| Tripal entity | cvterm | Chado table | Type (if not exclusive to the Chado table) |
|---|---|---|---|
| Stock | whole plant (PO:0000003) | stock | |
| Germplasm accession | Germplasm Accession (CO_010:0000044) | stock | type_id |
| Biological sample | biological sample (sep:00195) | biomaterial | |
| Analysis | Analysis (operation:2945) | analysis | |
| Genome annotation | operation:Genome Annotation | analysis | type_id analysisprop genome_annotation |
| Genome assembly | operation:Genome Assembly | analysis | type_id analysisprop genome_assembly |
| BLAST results | BLAST results (format:1333) | analysis | type_id analysisprop blast_analysis |
| InterPro results | InterPro results (local:interpro_results) | analysis | type_id analysisprop interpro_analysis |
| Project | Project (NCIT:C47885) | project | |
| Study | Study (SIO:001066) | study | |
| Gene | gene (SO:0000704) | feature | type_id |
| mRNA | mRNA (SO:0000234) | feature | type_id |
| Polypeptide | polypeptide (SO:0000104) | feature | type_id |
| QTL | QTL (SO:0000771) | feature | type_id |
| Genetic map | Genetic map (data:1278) | featuremap | type_id featuremapprop genetic |
| Physical map | Physical map (data:1280) | featuremap | type_id featuremapprop physical |

**Table 7.** Possible tables returned by the multi-omics API built in the ICGRC portal.

| Input rows | Output rows | Use case |
|---|---|---|
| cultivar, condition | trait | Field studies |
| marker, condition | trait | QTL mapping |
| cultivar, condition, tissue | transcripts, proteins, metabolites | Transcriptomics, proteomics, metabolomics studies |
| marker, tissue, condition | trait | GWAS |
| marker, tissue, condition | expression | eGWAS, eQTL |
| marker, tissue, condition | metabolites | mGWAS |
| transcripts | metabolites | Gene/metabolite association networks, |
| metabolites | traits | Multivariate studies |

result table should be able to handle any of the dataset types, factors, variables and values in Table 7.

The datasets are linked by sample IDs defined by their NCBI BioSample whenever available. Publications are required to define BioSample accessions (SAMNs) when they submit sequences to NCBI, including NGS genotyping and RNA-Seq studies.

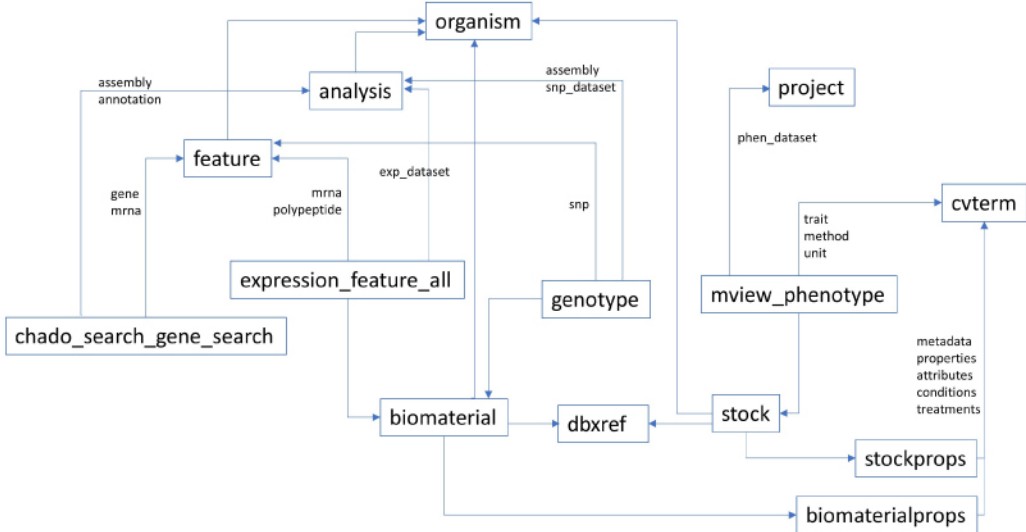

**Figure 3. Entity relationship diagram of Chado tables and materialized views involved in omics integration.**
The Tripal modules generate materialized views to aggregate data for genomic features and gene annotations, expression levels, and phenotype values. Genotype metadata on biomaterials and features are also in the database, while the allele values are in external vcf.gz files. Biomaterial and Stock are synchronized through dbxref using NCBI SRR or SAMN accessions when available.

**Table 8.** Definition of some Chado tables and Tripal BioEntities with ambiguous usage.

| Chado table | Tripal entity | Definition |
|---|---|---|
| Stock | | Any **stock** can be globally identified by the combination of organism, unique name and stock type. A **stock** is a physical entity, either living or preserved, held by collections. Stocks belong to a collection; they have IDs, type, organism, description and may have a genotype. |
| Biomaterial | | A **biomaterial** represents the MAGE concept of BioSource, BioSample, and LabeledExtract. It is essentially some **biological material** (tissue, cells, serum) that may have been processed. Processed biomaterials should be traceable back to raw biomaterials via the biomaterialrelationship table. |
| Organism | | The organismal taxonomic classification |
| | Genome assembly | Genome assembly is a type of **Analysis** |
| Study | | **Study** represents an experiment, published or otherwise, that produced microarray data |
| Project | | **Project** standard Chado flexible property table for projects. |
| Analysis | | An **Analysis** is a particular type of computational analysis; it may be a blast of one sequence against another, or an all-by-all blast, or a different kind of analysis altogether. It is a single unit of computation. |
| Assay | | An **Assay** consists of a physical instance of an array, combined with the conditions used to create the array (protocols, technician information). The assay can be thought of as a hybridization. |

We also identified several ambiguous usage of terminology while integrating the Tripal modules. Table 8 summarizes the exact definition of tables mentioned as defined in Chado, or BioEntities as defined in Tripal.

The 'stock' and 'biomaterial' tables appear to hold similar data. However, according to Chado's definition, 'stock' represents the physical entity of an organism identified by a unique name and stock type that is held within a living or preserved collection. In contrast,

'biomaterial' is a biological material (tissue, cells or serum) that may have been processed. Biomaterials should be traceable to source raw biomaterial, recognizing that unless vegetatively cloned, individual plants of a dioecious outcrossing species are likely to have different genetic makeup (haplotype). The Expression module uses the Biomaterial type, while the Phenotype module uses the Germplasm Accession type to represent the samples. Merging expression with phenotype data needs a link between stock and biomaterial, which is not explicitly defined in Chado. The workaround implemented in the project is to use the same property and value (NCBI BioSample ID) in the stockprop and biomaterialprop tables. However, SAMN IDs may not be available since only sequences submitted to NCBI or other INDSC repositories have them. In this situation, we assigned stocks/biomaterial IDs using the simplest and most unique name deduced from the source publication.

Issues in the 'organism' and 'genome assembly' entities were also encountered. Although the portal handles only *Cannabis sativa* L. as the target organism, the 'organism' entity is used in Chado and Tripal to define a set of contigs and genes for a given genome. Thus, the different genome assemblies for cs10, PK and FN are stored as distinct records in the organism table. In Tripal, the Genome Assembly entity is defined as a computational analysis for a genome assembly in the analysis table. As implemented in the project, we loaded the different assemblies as separate organisms in the organism table, then modified all Organism labels to Assembly.

For the entities 'Project', 'Study', 'Experiment' and 'Analysis', by Chado's definition, 'Study' represents an experiment that produced microarray data; 'Project' is a flexible property table for projects; 'Analysis' is a single unit of computational analysis; 'Assay' consists of a physical instance of an array which, when combined with specific conditions, creates an Array.

The different types of computational analysis are reflected in Tripal by using the specific analysis entities. Currently defined are Genome Annotation, Genome Assembly, BLAST Results and InterPro Results. The phenotype module loads the experiment (Experiment is used in the label) to the Project table and is associated with a genus and the samples to Germplasm Accessions. A genus is defined in the phenotype configuration to consist of a controlled vocabulary of Trait, Method and Unit, or optionally by Crop Ontology. 'Germplasm Accessions' is a type of stock in Tripal, so phenotype samples are loaded into the stock table. Phenotype studies are added to the Project table. The Expression and Biomaterials loader in the expression module associates the data to an Analysis and Organism, which was used earlier for assemblies.

The interchanging use of terms made it confusing to merge samples between different modules, and our workaround was to use common terms for both the stocksprops and biomaterialprops table.

The Tripal genotype module, which uses the Natural Diversity tables [94], was tested and found to be performing sub-optimally, especially when loading large genotyping data matrices generated from NGS variant calling. To solve the speed issue, the SNP-Seek software system [68], developed to host large SNP datasets, was instantiated in a podman container and embedded as a Tripal page. This feature, now named CannSeek database, is described in the accompanying paper. For the API calls to query SNPs, BCFtools (RRID:SCR_005227) [95] was used directly on the VCF files stored separately for each reference and dataset. The genotyped samples' metadata are loaded to the 'biomaterial' table so they can be autoloaded from NCBI BioSample.



Some issues for the entities 'gene', 'mRNA' and 'protein' were also resolved. The central dogma of molecular biology relations is well-defined in the Tripal model and all are stored in the Chado feature table by different feature types. Their relationships are explicitly related as: mRNA part_of Gene, Exon part_of mRNA, Polypeptide derives_from mRNA. BLAST and InterPro Analysis Results were assigned to the specific molecule type used (i.e., DNA, mRNA or polypeptide). In the 'chado_gene_search' module, the keyword search was modified to include the annotations assigned to all genes, mRNAs and proteins by tracing their relationships. However, it became challenging to cross-reference genes between external databases and applications. The accessions or IDs will be different for the same gene but different molecule types. In addition, a gene locus can be associated with multiple mRNAs and proteins. To avoid the complicated query, a materialized view was created to map the corresponding genes, mRNA and polypeptides.

Replicates are challenging to handle when samples are measured, analyzed, and reported on multiple data types. Replicates may be biological (i.e., multiple studies using the same plant source/genebank accession, multiple individuals in a study) or technical (i.e., multiple samples from a tissue with the same treatments). In NCBI, a biomaterial in BioSample can have multiple sequence replicates in SRA (SRR). Replicated NGS reads can be variant-called separately or jointly for a given SAMN. The measurement of metabolites for a SAMN can also have technical replicates. Some phenotype datasets loaded from publications report measurements per replicate [57], while others report mean and standard deviation [85]. The Tripal phenotype module documentation suggests loading the individual replicate data [94]. However, their proposed use case is focused on geolocational and seasonal factors, which may be more applicable for a single project or institute, but not for a community portal that integrates various data sources. These replicates create a Cartesian join between biomaterials and properties; that is, rows or columns will be duplicated except for one element. For this reason, we use average and standard deviation for technical replicates from published datasets. We pre-calculated these statistics before loading when only replicate values were available. It is also possible to have an array of values for a given biomaterial and property without a traceable replicate number. For publications that report in this format, the system can still store them and report in JSON format.

### The omics integration APIs: database query generation and web service implementation

The normalized structure of Chado makes database SQL queries complex and inefficient. To circumvent this, the Tripal modules prepare materialized views of the necessary data ready for user query and visualization. The same strategy was used to integrate the datasets from the different modules where they shared common BioEntities. Details of the data integration queries are described in (protocol step 16). In brief, five datatypes are queried separately, each returning four columns (i.e., datatype, property, sample and value). The data types are ID, PROP, PHEN, EXP and SNP. ID refers to sample identifications, PROP to sample attributes and metadata, PHEN are phenotype variables, EXP are transcripts or protein IDs, and SNPs are SNP IDs. The results are merged by a UNION operation, pivoted into a DATATYPE, PROPERTY by SAMPLE table, and returned to the client.

The API for omics data, implemented as a web service, usually returns JSON format text. They are returned for separate experiments, datasets, omics types like BrAPI [96] or Tripal

Web-Service [97]. Here, we return an efficient and intuitive table format ready for further processing, which can be loaded easily into spreadsheets or any csv reader library. This API will work in complement with the Tripal Web Service module, which is also enabled in the ICGRC site [98]. The web-services module can supply the metadata of the biological entities but not the values of the different omics measurements. Specifically, the values queried by this API are measurements in the materialized views taken from the Chado tables elementresult.signal (expression), phenotype.value (phenotype), stockprop.value (stock property values), biomaterialprops.value (biomaterial property values). The allele values for variant queries are read directly from the VCF files due to their large size if loaded to the Postgres database. The samples and metadata of the VCF are in the 'biomaterial' and 'biomaterialprop' tables.

To avoid Cartesian join duplication for replicates, complex element object representation, like JSON, was used to manage the list of values. Being able to present values with replicates, their statistics, methods and units as a table element, the table still has the equivalent representation power of a JSON object typical in web-services APIs.

## The Cannabis Tripal Web portal

The instantiated platform is available at [99]. Figure 4 shows the main features of the site. The functional menu includes Genomics, Phenome, Tools and Infome. 'Genomics' includes datasets pertaining to sequences, while 'Phenome' includes phenotypes and gene expression data. 'Infome' allows for the management of Tripal Bioentities metadata, manuals, and collaborative information. The 'Genetic Resources' section is reserved for data related to gene banks and other genetic resource collections. The 'Tools' section is for performing computations using the datasets; currently, only BLAST is available. Details on the use are provided in the User's Manual in the menu.

Drupal administrative modules allow site and user administration. Users are assigned roles of public, registered, data curator, site curator or administrator. Registered users can view additional information including the metadata of all samples, biomaterials, analyses, or any bioentities made private to a group of users. They can also post comments on some pages and data. To register, only a name, affiliation and business email address are required.

Most of the modules retain standard Tripal functionality, but some were revised to expand their capability. In 'Gene function search', keyword search now includes all annotation assigned to gene, mRNA or protein. Using a Gene Ontology accession (GO:nnnnnn) includes all sequence features annotated with any of the transitive closure of the GO term. The expression module has an added feature to generate a file intended for the Biological Pathway viewer. The 21-trichomes transcript expression levels were also converted to the bigwig format and visualized as JBrowse tracks at the coordinates identified with TransDecoder. Lastly, unique to this portal are the MapMan biological pathway expression viewer [90] and the CannSeek SNPs viewer. The gene search materialized view SQL definition and MapManJS HTML page revisions are in the protocols.io steps 11 and 18, respectively.

The protocol developed [46] can also serve for future site updates and in setting up new Tripal sites, especially for crops where only sequence datasets are readily available.

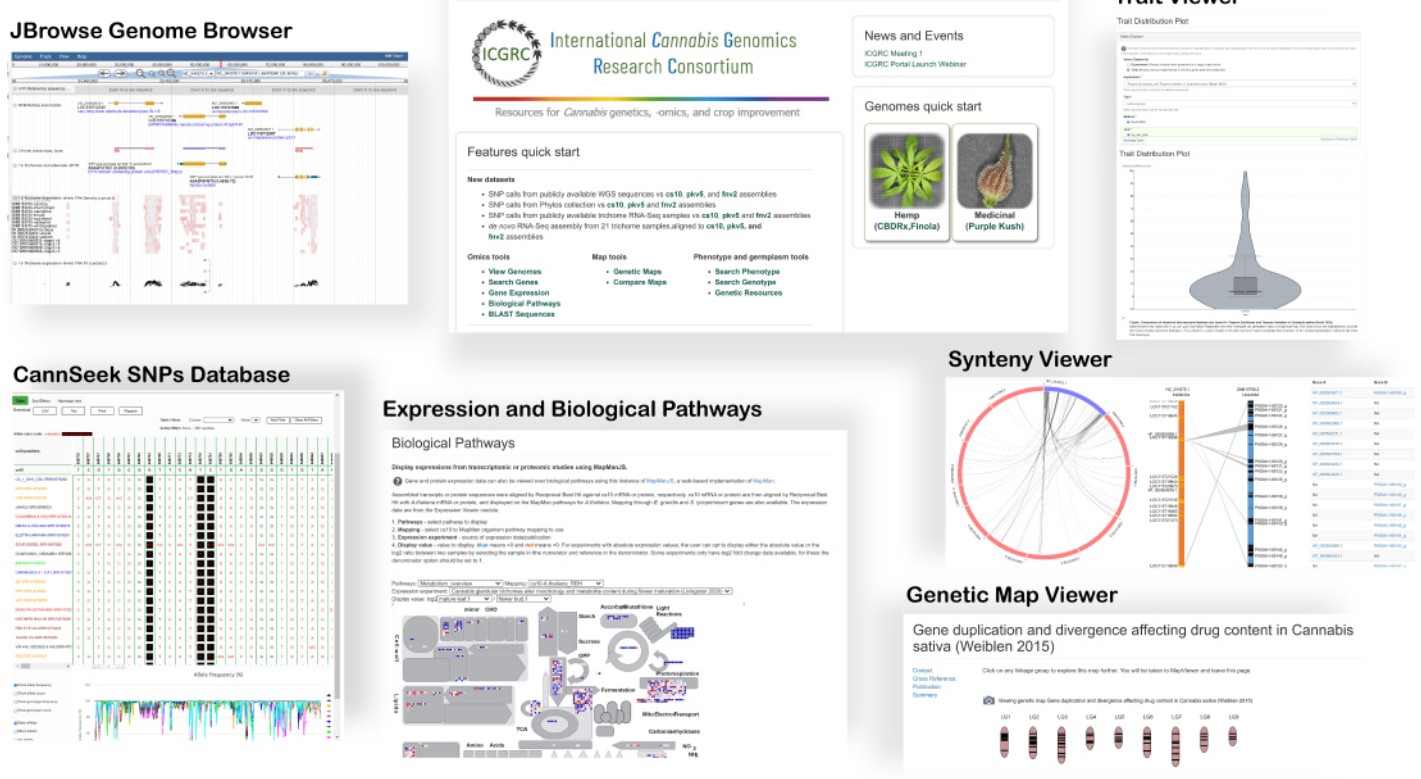

**Figure 4.** **ICGRC Cannabis Web Portal (icgrc.info) main feature pages.**
The web portal is a Tripal instance with various modules to host and visualize published cannabis omics datasets.

## The omics API query

We designed and implemented an API to serve programmatic queries to the omics datasets. The available queries and parameters are described in the ICGRC Omics API documentation [100]. The general API query format is, https://icgrc.info/api/{user}/{datatype_list}/{constraints_list}?parameters=..&..&... where {user} could be an authenticated user with login and privileges, or just 'user' for public access. {datatype_list} is a comma-separated set of datatypes to query with possible values of [gene, phenotype, expression, variant]. {constraints_list} is a comma-separated list of genomic regions [chromosome:start-end] or keywords. Genomic region is used when Variant is queried. Keywords are queried on the gene functional annotations, as used by the 'Gene Function Search' page, to get genes for Expression or Variant queries. The available datasets and summaries are available at https://icgrc.info/api/user/{datatype}/dataset. The results of queries with multiple datatypes are combined by SAMPLE. For example, for samples with both expression and phenotype data, the sample column will have both genes and traits/metabolite rows.



## DISCUSSION

## Standardization for cannabis vocabulary and nomenclatures

Standardization is mandatory to effectively use the platform when hosting diverse data types and sources. Here, we summarize the terms that require community-wide standardization, their impact on storage and analyses, the current best practices from various research groups, possible immediate workarounds, and suggestions to prompt the establishment of accepted community standards.

### *Sequence and sequence features accession IDs*

NCBI assigns unique sequence identifiers for all datasets it hosts and analyses. Similar ID assignments are implemented by other databases like Ensembl and UniProt. External references for these identifiers are handled in the 'dbxref' table in Chado.

However, prior to publication, different research groups tend to generate their own IDs and nomenclatures. It would be highly beneficial if a standard for generating nomenclatures were adopted by the community. The standard alphanumeric coding system may consider the increments, versioning, feature types, relationships between feature types, and relationships with other relevant ontologies.

### *Stocks, cultivars, biomaterials, samples*

This is a prevailing gene bank issue [101]; unfortunately, there is no authoritative gene bank for cannabis [102]. NCBI assigns sample SAMN IDs for submissions in BioSample. However, these IDs are more applicable for biomaterials, specifically laboratory materials. They cannot trace the pedigrees of genetic stocks.

In the absence of a physical gene bank, the creation of a virtual gene bank was suggested [36]. A possible model for this is the Genesys platform [103]. Under this system, physical gene banks worldwide publish their passport and characterization data, which are then merged by Genesys into a common query interface. This or a similar entity would then have the authority to assign Stock or Germplasm accessions for cannabis.

### *NCBI BioSample metadata submissions*

Related to biomaterials, the NCBI BioSample SAMN metadata can be loaded automatically into Tripal. Thus, terms used in NCBI submissions are reflected in the Tripal 'Biomaterial' pages. The sample metadata are also in the ID, and PROP rows are returned to API queries. Therefore, it is beneficial to use standardized terminologies during submissions to public repositories, like NCBI, to use the information efficiently and accurately in subsequent analyses.

Querying PubMed for publications relating to cannabis plant biology or agronomy is complicated by a wealth of papers pertaining to clinical or pharmacological studies. We therefore recommend using standard keywords or MESH terms specific to the study of the plant itself, not its clinical/pharmacological use.

### *Traits, variables, methods and units for cannabis*

Detailed crop ontologies are available for well-studied crops, although these have their limitations [104]. These terms are often derived from legacy dictionaries used by gene banks in their passport transactions and characterization of collections. A standardized cannabis crop ontology would benefit both the industry and scientific community in

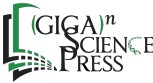

facilitating data curation and management. Moreover, using them during submission to public repositories simplifies their integration, as mentioned above.

### Site administration

The site is hosted and maintained by the Southern Cross University in coordination with the ICGRC steering group to appoint a curation team. To maintain data integrity, ICGRC will, as much as possible, load only data that have been peer-reviewed and published. For unpublished data that are submitted using the website feature, these need to be reviewed by the curating team. Datasets that are evaluated to be important to add to the database may undergo secondary analyses, and results may be added to the portal. Newly submitted data, after vetting and curation, will be incorporated into the yearly cycle update of the site.

Users can register to be authenticated using their business email, but those using free email service for their business may contact the site administrator to establish their bona fide credentials. Authenticated users can post comments and participate in collaborative discussions.

### Multi-omics query using the API: use-cases

With the different omics data types loaded in the database, we are able to integrate these using API queries. We created Jupyter (RRID:SCR_018315) notebooks and Python modules to demonstrate the potential uses of the API for multi-omics analyses, all available in (protocol step 17). A static export of the notebook with outputs is in the ICGRC Omics Demo [105]. In the notebook, the Python modules (module names are in `courier font`) perform specific tasks and use Pandas (RRID:SCR_018214) Dataframe for data handling and manipulation. Depending on the intended analysis, additional software or modules need to be installed in the client as described in the notebook.

### *Use case 1*

Get phenotypes, normalize units, plot distributions and batch effects correction
https://icgrc.info/api/user/phenotype/list.

This query returns a table of phenotype values from multiple datasets in mixed units and format. When merging values from multiple sources, compatible variables need conversion of units. Units and standard deviation information for each value are included with flag with_stdunits=1. The values with different units can then be converted to a uniform unit (`convert_units_to`) for plotting and further processing. The main purpose of this feature is storage, integration and retrieval. For the statistical validity of merging data from different sources and studies, we included a minimal feature (`check_batcheffects`) to detect and correct for batch effects and impute missing values using the samples' metadata. This module uses ski-learn SimpleImputer and PCA [106], ppca [107] and pyComBat (RRID:SCR_010974) [108]. The metadata and sample attributes can also be used as covariates in various analyses.

### *Use case 2*

Co-expression analysis using WGCNA
https://icgrc.info/api/user/expression,phenotype/list.

This query returns a table with phenotype and gene expression values for each sample in the selected datasets. These data are used in weighted correlation network analysis, or



WGCNA (RRID:SCR_003302) [109], in the (`do_wgcna_prepare`) and (`do_wgcna_modulesgenes`) modules. These modules return the top gene-trait pairs with gene significance and *p*-value.

### Use case 3

Matrix eQTL

https://icgrc.info/api/user/expression,variant/{querygenes}.

This query returns a table of gene expressions and SNP allele data for genes and regions that meet the querygenes annotations constraints. The module (`do_matrixeqtl`) uses Matrix eQTL [110] for the top SNP-gene pairs by *p*-value. If provided with SNP and gene position information, Matrix eQTL can also analyze for local (cis) and distant (trans) pairs separately. The gene information is also queried using the API https://icgrc.info/api/user/gene, constrained by position for the SNPs or by accession for the gene expression set.

### Use case 4

SNP-trait association study

https://icgrc.info/api/user/variant,phenotype/{querygenes}.

This query returns a table of traits, mostly metabolite contents for cannabis, and SNP alleles for genes and regions in querygenes. The module (`do_gwas`) uses PLINK1 (RRID:SCR_001757) [111] –assoc function to get the top SNP-phenotype associations by *p*-value. When performed on all SNPs for a genome and with enough samples, this is basically a genome-wide association study (GWAS). To date, there are only a few public cannabis samples that have both phenotype and genotype data publicly available, hence the limitation on datasets loaded in the portal.

The omics tools presented above used the API to retrieve multiple datasets for various analyses. Each analysis returns the top associations by *p*-value between genes, genotype variants, biomolecules, traits or other factors. From the multiple evidence collected, the logical next step is to overlap the relationships into networks (union) or Venn diagrams (intersection) for further interpretation. The merge module (`merge_matrixeqtl_wgcna_gwas`) takes the results from WGCNA, eQTL and plink -assoc, intersects the genes and traits, and merges the pairs to identify SNP gene-traits associations supported by two pieces of evidence, as illustrated in Figure 5.

The example use cases illustrate the flexibility of the API for multi-omics and multi-source analyses. We showed two ways of merging multiple datasets. First, datasets are merged as input for an analysis tool. This requires shared samples, batch effect correction, covariates information, and most likely heavy imputations. Their essential link is the shared genotype. In the second approach, the input for an analysis tool is from one dataset, but the results from different analyses are merged by shared biomolecules. Each approach has its advantages and applications. Integration of sequence data is performed in the backend during variant calling, transcript assembly, or gene expression quantification.

## CONCLUSION

We set up the Tripal platform to host cannabis genomics data that is available in the public domain. We also provided a protocol to generate various datasets from sequence data, to load and test the different modules and visualizations. The protocol simplifies the process of regularly updating the site. One year after our last upload, a recent check on NCBI and

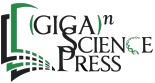

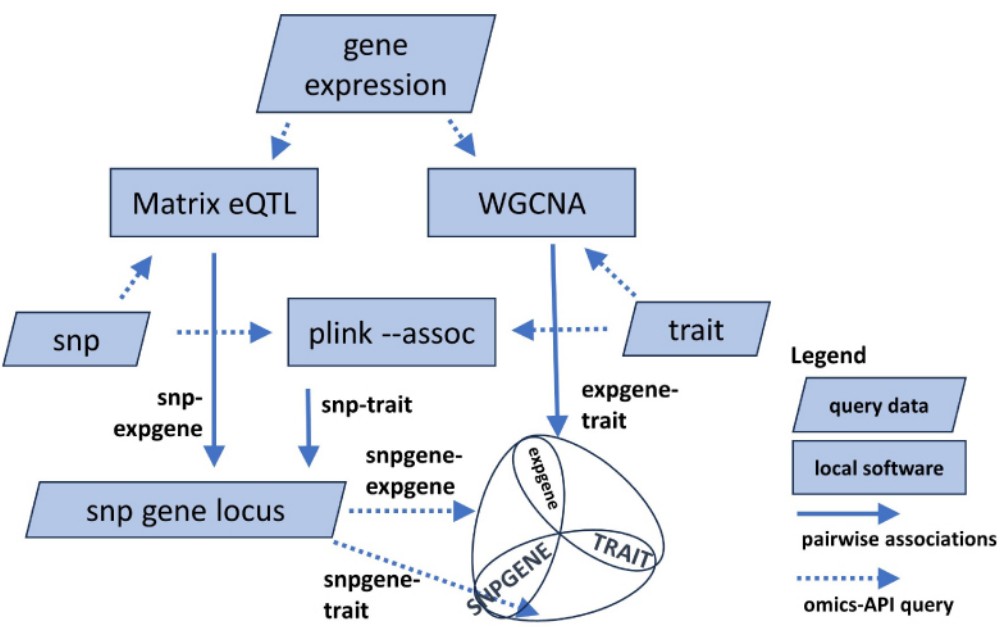

**Figure 5.** **Use cases on multi-omics analyses using local software and the ICGRC omics API.**
The different use cases in the Jupyter notebook demonstration can be found at [105]. Results from the analysis are the top associations of SNP-transcript (Matrix eQTL), transcript-trait (WGCNA), or SNP-trait (plink -assoc). The common datatypes between tools are SNPs (Matrix eQTL-plink), transcripts (Matrix eQTL-WGCNA), or traits (plink-WGCNA). Combining their results generates a list of candidate genes with two independent evidence of associations.

PubMed returned new publications and datasets for potential reanalysis and loading to the site. This shows that cannabis research is active and rapidly growing. The setup of this web portal is timely and ready to serve the research community. We highlight the importance of using community standards, especially when depositing datasets in public repositories. Standards minimize third-party curation and encourage reuse for meta-analyses. The multi-omics integration API and the example Jupyter notebooks demonstrate the convenience of doing analysis when the data are readily available in table format. Since they are built on top of Tripal modules, they can be readily adapted to any Tripal site for other research communities.

## AVAILABILITY OF SOURCE CODE AND REQUIREMENTS

The source codes in this project can be grouped into:

(a) The ICGRC web application, which is an instance of Tripal v3, the various Tripal/Drupal modules installed, site customizations/settings saved in the database, and the custom omics API web-server-module developed in this project. These are available Files in [112]
(b) the scripts and command lines part of the computational protocol for dataset generation from public datasets
(c) Jupyter notebooks and Python client library for omics analyses, which call the omics API.

(b) and (c) are embedded or attached in the sections of the protocol described below

- Project name: **ICGRC Portal Tripal Data Generation and Setup**
- Project home page: https://www.protocols.io/view/icgrc-portal-tripal-data-generation-and-setup-n2bvj3nz5lk5
- Operating system(s): Linux
- Programming language: bash, PHP, Python
- Other requirements: listed in the steps in the protocol
- License: CC BY 4.0 International
- DOI: https://doi.org/10.17504/protocols.io.n2bvj3nz5lk5/v3

- Project name: **ICGRC web-portal**
- Operating system(s): Linux
- Programming language: bash, PHP, SQL
- Other requirements: containerization via podman or Docker
- License: GNU General Public License 2.0
- DOI: https://doi.org/10.5524/102591

The Tripal web server runs as a podman (or docker) container using an image pulled from **docker pull statonlab/tripal3**. For the ICGRC customized site, the Postgres database and Apache server HTML directories were mapped into host server volumes. Various Drupal and Tripal modules were installed and enabled. Various datasets were either downloaded or generated from publications, and then loaded into the Tripal Postgres database.

Customized site Tripal modules, podman container and populated Postgres database are available in GigaDB [112].

- Project name: **ICGRC -Omics API**
- Project home page: https://www.protocols.io/view/icgrc-portal-tripal-data-generation-and-setup-n2bvj3nz5lk5?step=17
- Operating system(s): Linux
- Programming language: bash, Jupyter, Python, R
- Other requirements: listed in specific modules in the demonstration notebook https://snp.icgrc.info/static/icgrc_omics_demo.html
- License: CC BY 4.0 International
- DOI: https://doi.org/10.17504/protocols.io.n2bvj3nz5lk5/v3

Availability of Jupyter notebooks, Python modules, and sample inputs are described in the protocol.io [46] (Figure 6).

## DATA AVAILABILITY

The intermediate files generated by the steps in [46] ready for loading to the Tripal database are available in Files in [112]. The files are labeled following the step number in protocols.io. These files are to be loaded into Tripal using the various modules' data-loaders web-interface. These files include:

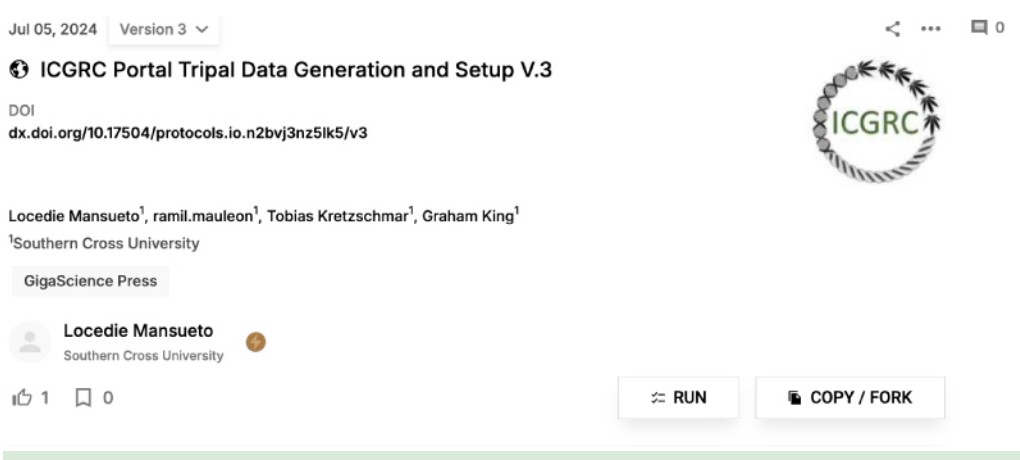

**Figure 6.** Protocols.io protocol for the ICGRC Portal Tripal Data Generation and Setup [46]. https://www.protocols.io/widgets/doi?uri=dx.doi.org/10.17504/protocols.io.n2bvj3nz5lk5/v3

- Flat files in a format accepted by Tripal:
  - PK and FN gene models gff, cds, protein fasta, blast interpro results
  - 21TRICH transcript fasta, gff, expression levels
  - Biopathway-cs10 alignments (attached in protocols)
  - Synteny blocks (blocks tsv)
  - Phenotype datasets (curated/formatted from publications)
  - GEO (curated/formatted from publications)
  - Maps, QTL (curated/formatted from publications)

- Tripal web server Docker image with internal paths pointing to external volumes and start-up script with volumes mapping specifications.
- The archived directories to be mapped as Docker/podman volumes for the (a) Postgres data, (b) Tripal customized site and modules, and c. directories hosting flatfile datasets and utilities. The Postgres database is preloaded with the current snapshot of the website.

## ABBREVIATIONS

API, Application Programming Interface; CBD, cannabidiol; cs10, CBDRx; FN, Finola; GEO, Gene Expression Omnibus; GWAS, genome-wide association study; ICGRC, International *Cannabis* Genomics Research Consortium; NCBI, National Center for Biotechnology Information; NGS, Next Generation Sequencing; PK, Purple Kush; QTL, quantitative trait loci; RefSeq, Reference Sequence Database; RNA-Seq, RNA sequencing; SAMN, BioSample accession; SNP, single nucleotide polymorphisms; SRA, Sequence Read Archive; SRR, sequence replicates in SRA.

## DECLARATIONS

### Ethical approval

The authors declare that ethical approval was not required for this type of research.

### Competing interests

The authors declare that they have no competing interests.

## Author contributions

LM implemented and modified the architecture and contents of the software and is the primary author of the manuscript. RM supervised the software development and contributed to the development of the manuscript. TK supervised the overall development of the manuscript as project leader. GJK contributed to the development of the manuscript and provided linkage to the International Cannabis Genomics Research Consortium. All authors have proofread the manuscript.

## Funding

This study was funded by the Australian Research Council (ARC) Linkage project LP210200606. In addition, first author LM received a stipend from Southern Cross University (SCU). The ICGRC and CannSeek web servers are hosted and funded by SCU.

## Acknowledgements

The authors acknowledge the provision of computing and data resources provided by the Australian BioCommons Leadership Share (ABLeS) program. This program is co-funded by Bioplatforms Australia (enabled by NCRIS), the National Computational Infrastructure and Pawsey Supercomputing Centre.

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
