## [Editor Report]

Editor’s AssessmentThis paper reports the establishment of the International Cannabis Genomics Research Consortium (ICGRC) web portal leveraging the open source Tripal platform to enhance data accessibility and integration for Cannabis sativa (Cannabis) multi-omics research. With the aim of bringing together the wealth of publicly available genomic, transcriptomic, proteomic, and metabolomic data sets to improve cannabis for food, fiber and medicinal traits. Tripal is a content management system for genomics data, presenting a ready-to-use specialized ‘omics modules for loading, visualization, and analysis, and is GMOD (Generic Model Organism Database) standards-compliant. The paper explaining how this was put together, what data and features are available, and providing a case study for other communities wanting to create their own Tripal platform. Covering their setup and customizations of the Tripal platform, and how they re-engineered modules for multi-omics data integration, and addition of many other custom features that can be reused. Peer review fixed a few minor bugs and added clarifications on how the platform will be updated.Editor’s AssessmentThis paper reports the establishment of the International Cannabis Genomics Research Consortium (ICGRC) web portal leveraging the open source Tripal platform to enhance data accessibility and integration for Cannabis sativa (Cannabis) multi-omics research. With the aim of bringing together the wealth of publicly available genomic, transcriptomic, proteomic, and metabolomic data sets to improve cannabis for food, fiber and medicinal traits. Tripal is a content management system for genomics data, presenting a ready-to-use specialized ‘omics modules for loading, visualization, and analysis, and is GMOD (Generic Model Organism Database) standards-compliant. The paper explaining how this was put together, what data and features are available, and providing a case study for other communities wanting to create their own Tripal platform. Covering their setup and customizations of the Tripal platform, and how they re-engineered modules for multi-omics data integration, and addition of many other custom features that can be reused. Peer review fixed a few minor bugs and added clarifications on how the platform will be updated.

---

## [Reviewer Report]

Reviewer name and names of any other individual's who aided in reviewerWeiwen WangDo you understand and agree to our policy of having open and named reviews, and having your review included with the published manuscript. (If no, please inform the editor that you cannot review this manuscript.)YesIs the language of sufficient quality?YesPlease add additional comments on language quality to clarify if neededIs there a clear statement of need explaining what problems the software is designed to solve and who the target audience is? YesAdditional CommentsIs the source code available, and has an appropriate Open Source Initiative license <a href="https://opensource.org/licenses" target="_blank">(https://opensource.org/licenses)</a> been assigned to the code?YesAdditional CommentsAs Open Source Software are there guidelines on how to contribute, report issues or seek support on the code?YesAdditional CommentsIs the code executable?Unable to testAdditional CommentsThis manuscript is about an online platform, and I am not sure how to test the code.Is installation/deployment sufficiently outlined in the paper and documentation, and does it proceed as outlined?Unable to testAdditional CommentsSame as above.Is the documentation provided clear and user friendly?YesAdditional CommentsIs there enough clear information in the documentation to install, run and test this tool, including information on where to seek help if required?Additional CommentsIs there a clearly-stated list of dependencies, and is the core functionality of the software documented to a satisfactory level?YesAdditional CommentsHave any claims of performance been sufficiently tested and compared to other commonly-used packages? Not applicableAdditional CommentsIs test data available, either included with the submission or openly available via cited third party sources (e.g. accession numbers, data DOIs)?Additional CommentsAre there (ideally real world) examples demonstrating use of the software? YesAdditional CommentsIs automated testing used or are there manual steps described so that the functionality of the software can be verified?Additional CommentsAny Additional Overall Comments to the AuthorWith the increasing legalization of cannabis in many countries today, exploring this crop has become a hot topic of research. This manuscript by Mansueto et al. introduces a platform built on the Tripal framework, designed to facilitate multi-omics research in Cannabis sativa. The platform integrates genomic, transcriptomic, proteomic, and metabolomic data, providing researchers with a comprehensive resource for data analysis and sharing. Additionally, APIs have been developed, enabling rapid querying. This manuscript detailed information on how to customize Tripal modules and Chado schema for managing biological entities. Finally, this manuscript highlights the importance of standardization in data storage and analysis, proposing community-wide adoption of standardized nomenclature to ensure consistency and traceability of data. Overall, the platform is poised to become a valuable resource for cannabis research and to advance scientific progress in related fields. While this manuscript was engaging, particularly in the sections on Tripal "re-engineering" and controlled vocabulary, I do have several concerns. 1 Because my registration (using business email) has not been approved, I cannot test the functions requiring ICGRC registration. 2 The authors noted that the Cannabis Genome Browser has not been updated. Do the authors have a plan for updating ICGRC? If so, what is the proposed update frequency? 3 ICGRC currently includes only a few cannabis cultivars, especially when compared to other platforms like Kannapedia and CannabisGDB. Do the authors have plans to add additional cultivars, such as First Light and Jamaican Lions mentioned in this manuscript, in the near future? 4 When I tried to register using Gmail, an error popped up: ‘Domain is not allowed to register for this site’. Perhaps it would be clearer to instruct users to use a business email for registration directly. 5 There is a data submission function in ICGRC, but the exact workings of this feature remain unclear to me. If a user submits a cannabis genome to the ICGRC, whether this data will be visualized within specific modules like synteny search or genetic mapping tools on the platform.RecommendationMinor Revisions

---

## [Reviewer Report]

Reviewer name and names of any other individual's who aided in reviewerhongyun shangDo you understand and agree to our policy of having open and named reviews, and having your review included with the published manuscript. (If no, please inform the editor that you cannot review this manuscript.)YesIs the language of sufficient quality?YesPlease add additional comments on language quality to clarify if neededIs there a clear statement of need explaining what problems the software is designed to solve and who the target audience is? YesAdditional CommentsIs the source code available, and has an appropriate Open Source Initiative license <a href="https://opensource.org/licenses" target="_blank">(https://opensource.org/licenses)</a> been assigned to the code?YesAdditional CommentsAs Open Source Software are there guidelines on how to contribute, report issues or seek support on the code?YesAdditional CommentsIs the code executable?Unable to testAdditional CommentsIs installation/deployment sufficiently outlined in the paper and documentation, and does it proceed as outlined?Unable to testAdditional CommentsIs the documentation provided clear and user friendly?YesAdditional CommentsIs there enough clear information in the documentation to install, run and test this tool, including information on where to seek help if required?Additional CommentsIs there a clearly-stated list of dependencies, and is the core functionality of the software documented to a satisfactory level?YesAdditional CommentsHave any claims of performance been sufficiently tested and compared to other commonly-used packages? YesAdditional CommentsIs test data available, either included with the submission or openly available via cited third party sources (e.g. accession numbers, data DOIs)?Additional CommentsAre there (ideally real world) examples demonstrating use of the software? YesAdditional CommentsIs automated testing used or are there manual steps described so that the functionality of the software can be verified?Additional CommentsAny Additional Overall Comments to the AuthorThis is a comprehensive database with many features that improves the shortcomings of cannabis species that had no genome database in the past. It is a good work. Here are some minor suggestions: 1. Did not find the function of searching gene and protein sequences directly by gene id without providing chromosome location, which may be a common feature of many omics databases. 2. In the chapter "The need for cannabis multi-omics databases and analysis platforms", "There are no analysis tools or results available on this website", "No results available" seems inappropriate. 3. In the chapter "Cannabis - Omics, Genetic and Phenotypic Datasets in the Public Domain", "Crop Ontology" Crop Ontology, is "Crop Ontology" repeated?RecommendationMinor Revisions